# Increasing the Efficiency of the Accumulation of Recombinant Proteins in Plant Cells: The Role of Transport Signal Peptides

**DOI:** 10.3390/plants11192561

**Published:** 2022-09-28

**Authors:** Sergey M. Rozov, Elena V. Deineko

**Affiliations:** Federal Research Center Institute of Cytology and Genetics, Siberian Branch of Russian Academy of Sciences, Pr. Akad. Lavrentieva 10, Novosibirsk 630090, Russia

**Keywords:** recombinant protein, transport signal peptide, plant expression system

## Abstract

The problem with increasing the yield of recombinant proteins is resolvable using different approaches, including the transport of a target protein to cell compartments with a low protease activity. In the cell, protein targeting involves short-signal peptide sequences recognized by intracellular protein transport systems. The main systems of the protein transport across membranes of the endoplasmic reticulum and endosymbiotic organelles are reviewed here, as are the major types and structure of the signal sequences targeting proteins to the endoplasmic reticulum and its derivatives, to plastids, and to mitochondria. The role of protein targeting to certain cell organelles depending on specific features of recombinant proteins and the effect of this targeting on the protein yield are discussed, in addition to the main directions of the search for signal sequences based on their primary structure. This knowledge makes it possible not only to predict a protein localization in the cell but also to reveal the most efficient sequences with potential biotechnological utility.

## 1. Introduction

The latest advances in molecular biology and genetic engineering allow for the production of a large number of diverse recombinant proteins for medical applications and vaccines—for example, antibodies and commercial enzymes—using different expression platforms [1,2,3]. These proteins are synthesized with the help of prokaryotic and eukaryotic expression systems, such as *Escherichia coli,* yeasts, and cultured cells of insects and mammals. Over half of pharmaceutically valuable proteins are produced in mammalian cells [2,3] because these cells provide correct post-translational modifications (PTMs) of the eukaryotic type [4,5]. Incorrect PTMs—for example, in yeasts—or their absence in prokaryotes may worsen the properties of a synthesized recombinant protein—in particular, by altering its biological activity and pharmacokinetics. That is why prokaryotic expression systems are mainly used for synthesizing relatively simple therapeutic proteins. As for more complex proteins, they are usually synthesized in expression systems involving mammalian cell cultures [5]; however, these systems also have their flaws. First and foremost, there is a high cost of cultivation, difficulties with the upscaling of the processes, and a potential virus infection. Plant cell cultures are free of these shortcomings, thereby representing the most promising alternative for the synthesis of recombinant proteins. Plant cells combine the ability to perform PTMs according to the eukaryotic pattern and to rapidly grow, being both simple and inexpensive, similar to bacterial expression systems. In addition, plant systems exclude any potential contamination by animal viruses and bacterial toxins [2,6,7]. These advantages have formed the basis for so-called green pharmaceutics, in which plants or plant cell cultures act as biofactories producing recombinant proteins [7,8]. Unfortunately, plant expression systems are not free of certain flaws either; the main one is a low yield of recombinant proteins despite all attempts to increase it [9,10]. Several reviews describe the different methods used for enhancing the biosynthesis of transgene-encoded proteins [10,11,12].

The plant cell, as a representative of eukaryotes, has an extremely intricate pattern of intracellular compartmentalization and contains a complex system of subcellular membranes. When considering the plant cell in terms of an experimental tool for the biosynthesis of recombinant proteins, it should be kept in mind that the final yield of a target recombinant protein is determined not only by the intensity of its biosynthesis in the cytosol but also by the rate of its degradation in a given cell compartment. The cytosol has a high protease activity; moreover, in the cytosol, proteins can undergo undesirable modifications altering their folding [13]. However, all improperly folded proteins are degraded via the ubiquitin–proteasome proteolytic pathway [14], which leads to an even greater drop in the yield of a recombinant protein. Usually, the accumulation of recombinant proteins in the cytosol is rather low [15]; accordingly, this compartment is regarded as unsuitable for the storage of target recombinant proteins. The difference in the number of accumulated recombinant proteins between the cytosol and apoplast may reach several-thousand-fold [16]. Thus, it is reasonable to immediately transport the synthesized proteins to the apoplast or some other cell compartments. Cell organelles contain different types and numbers of proteases, and various recombinant proteins are, to various degrees, prone to degradation by individual proteases; the lowest protease activity is present in the apoplast and plastids [17,18]. Therefore, the selection of a cell compartment for the accumulation of a specific recombinant protein is not subject to simple universal rules.

Most of proteins in the plant cell are synthesized by the ribosomes in the cytosol, including secreted proteins and proteins of cell organelles (mitochondria, the nucleus, and chloroplasts). That is why the cell, during its evolution, has acquired the efficient mechanisms for transporting newly synthesized proteins to suitable cell organelles and even through the endoplasmic reticulum (ER) and Golgi apparatus to vacuoles and/or further to the apoplast. Specialized signal peptide (SP) sequences with a length of 20–80 amino acid residues (aa) are used for the transport of proteins to the target compartments; most frequently, these signal sequences are localized to the N terminus of proteins and, as a rule, are cleaved by specific proteases after reaching the target compartment [19]. For example, over 3000 cytosol-synthesized proteins of different types are targeted to chloroplasts by transit peptides [20,21]. These peptide signal sequences can be successfully employed to target recombinant proteins synthesized in plant cells to the compartments with a low protease activity. For this purpose, researchers use transfection with genetic constructs carrying the gene of a target protein fused in a head-to-tail manner to the sequence coding for the corresponding signal transport peptide [19,22,23]. This subcellular addressing of recombinant proteins is key to the boosting of protein accumulation because the cell compartment housing proteins directly influences their folding, assembly, and PTMs and prevents the direct degradation and interference of the proteins with the cell metabolism, which often takes place in the cytosol. The attachment of SPs for the transport of recombinant proteins to “protected” cell compartments or the apoplast usually guarantees higher levels of their production. In addition, some proteins may be cytotoxic in the cytosol but nontoxic when shielded by the double membrane of the chloroplast [16,24]. The apoplast, ER, and chloroplasts [22,23], as well as storage protein bodies (PBs), characteristic for plant seeds [25], are the most promising compartments for the accumulation of recombinant proteins. The SPs targeting different proteins to the same organelles differ not only in their amino acid sequences but also in the efficiency of delivery; accordingly, the yield of recombinant proteins carrying different SPs can vary considerably [23,26,27]. Thus, the expression efficiency of recombinant proteins in plant systems depends, to a considerable degree, on the selection of their subcellular targeting and the most efficient transport SPs.

## 2. Intracellular Transport of Proteins to the ER and Its Derivatives

Eukaryotes possess two types of ribosomes, namely, heavy 80S ribosomes (residing in the cytosol), lighter 70S ribosomes (residing in the endosymbiotic organelles), mitochondria, and plastids in plants. Nonetheless, 99% of mitochondrial proteins [28] and 95% of plastid proteins [29] are encoded by nuclear genes and synthesized on ribosomes of the cytosol. In addition, over one-third of all proteins synthesized by the eukaryotic cell have to, in part of in full, start making their way (to a destination) in the ER, where they undergo PTMs, are folded and verified for correct conformation, interact with chaperones, and form complexes of subunits [30,31,32,33]. The overall intracellular transport of proteins in the plant cell falls into two major routes, the first one comprising the translocation of proteins to the ER and its derivatives (Golgi complex, vacuoles, PBs, and apoplast). The second route includes the translocation of proteins to the nucleus and endosymbiotic organelles (mitochondria and plastids). Commonly, two types of protein transport to all eukaryotic cell compartments are distinguished, namely, cotranslational and post-translational. In the former, a protein is synthesized directly on the membrane of the ER or an organelle and cotranslationally transferred through the membrane; in the latter, the protein completely synthesized in the cytosol is delivered one way or another to its destination. It is not entirely clear what determines the choice of a pathway.

### 2.1. Cotranslational Protein Targeting to the ER and Its Derivatives

Many proteins targeted to the ER follow the signal recognition particle (SRP) route. The SRP is an RNA–protein complex of 7S RNA and six small proteins [34]. As a rule, proteins targeted via the SRP pathway carry either a hydrophobic SP at their N terminus, which is subsequently cleaved by a specific signal peptidase (SPase) in the ER, or an internal uncleaved SP acting as a signal anchor [35,36]. The cotranslational protein transport commences with the SRP binding to a ribosome with the subsequent scanning of the nascent chain for the presence of signal sequences. As soon as the SP or signal anchor leaves the ribosome, it binds to the SRP and activates it [37]. The activated SRP, together with the ribosome, the mRNA, and a nascent polypeptide chain (ribosome–nascent chain complex, RNC), temporarily arrests translation to perform the GTP-dependent translocation of the entire complex to the cognate SRP receptor on the outer ER side. The interaction with the receptor weakens the association of the SRP with the RNC, and the ribosome with mRNA and the nascent polypeptide chain remains on the outer ER membrane, where it interacts with heterotrimeric translocon Sec61. After this event, the translation restarts, and the synthesized protein is gradually translocated across the Sec61 transmembrane channel into the ER lumen (Figure 1) [35,36,38].

Several additional components involved in the maturation of newly synthesized proteins and the implementation of PTMs and chaperone-like folding functions are associated with the Sec61 translocon. One of these components is the translocon-associated protein complex (TRAP). The TRAP is a heterotetrameric complex of membrane proteins [39], assists Sec61 with protein translocation across the membrane, and controls the correct protein topogenesis in the ER lumen [40]. Another extra component of the Sec61 translocon is the oligosaccharyltransferase (OST) complex, which catalyzes protein N-glycosylation. This is a multisubunit complex comprising at least seven membrane proteins [41].

### 2.2. Post-Translational Targeting to the ER

Unlike cotranslational protein targeting, post-translational targeting commences only after a protein is completely synthesized and leaves the ribosome for the cytosol. Important specific features of post-translationally translocated polypeptides are their relatively small size (not exceeding 100 aa) and the C-terminal position of the transmembrane helix (TMH), acting as a translocation SP or, as in the case of yeasts, a “weak” N-terminal SP required for targeting [42]. These specific features interfere with the effective recognition of these proteins by the cotranslational targeting factor SRP and require the presence of a post-translational targeting factor—for example, GET3 (GET, guided entry of tail-anchored proteins). GET3 is involved in the targeting of tail-anchored (TA) proteins, which carry one region of the TMH (at the C terminus), acting as a guiding signal. In addition, GET3 enhances the targeting of short polypeptides with a cleavable N-terminal SP [43,44,45]. Two factors—SGTA [46] and heterotrimeric BAG6 [47]—participate in the translocation of a targeted protein to the GET3 complex [48]. Recently, one more SRP-independent targeting route, SND, was discovered; presumably, it acts as a reserve system for the translocation of certain protein types and overlaps with GET and SRP routes [49,50]. Many cytosol chaperones enhance post-translational protein targeting [51,52].

### 2.3. Structure of SPs Targeting Proteins to the ER and Its Derivatives

The SPs of different proteins, mediating their delivery to the ER and its derivatives, considerably differ in their primary structure but have a common structural design. They consist of 20–30 aa and comprise three domains: a positively charged N-terminal domain (1–7 aa), a central hydrophobic domain (7–15 aa), and a polar C-terminal one (3–7 aa), carrying an SPase recognition site [53]. Longer SPs are more abundant in eukaryotes but are also detectable in viral and bacterial proteins. Often, long SPs are not cleaved during protein maturation and may perform some other fractions along with targeting [54].

The N-terminal domain is responsible for the interaction with negatively charged phosphate groups of the double lipid membrane and is important for the protein translocation across it [55] and for the interaction with the phosphate backbone of the SRP complex [56]. A positive charge of the N-terminal domain also determines the protein orientation favorable for its translocation across the membrane [57]. The high hydrophobicity of the central SP domain results from a large number of leucine residues, which stabilize its α-helical structure [57,58]. The hydrophobic domain determines the SP orientation relative to the ER membrane, subsequent processing (for example, N-glycosylation), and the rate and efficiency of protein translocation across the membrane. A decrease in the hydrophobicity of the domain lowers the rate of protein translocation to the ER [59]. The helix-breaker residues Gly, Pro, and Ser, situated in the central part of the hydrophobic domain, give rise to a hairpin-like structure, which enhances SP insertion into the membrane and its subsequent cleavage by SPase [60]. A change in the position of helix-breaker residues in the hydrophobic domain may substantially change the level of protein accumulation [61]. In particular, the insertion of Gly into the center of the hydrophobic SP domain of recombinant cyclodextrin glucanotransferase doubles the level of protein secretion [62]. The C-terminal domain of SPs of different eukaryotic proteins somewhat varies in its length (3–7 aa) and amino acid composition and is composed of neutral or negatively charged amino acids that form a β-sheet, representing an SPase recognition site. This site is rather conserved among almost all translocated eukaryotic proteins but considerably differs among only a few lipoproteins. Positions −1 and −3 from the cleavage site (most frequently containing an Ala residue and forming the so-called AXA motif) are of paramount importance. A substitution with another amino acid, especially at position −1, causes a shift in the cleavage site [53,63] (Figure 2).

Gly, Ser, and Cys occur infrequently at positions −1 and −3 [64], whereas Ala, at position −3, can be replaced with Val, Leu, or Ile [65]. The amino acid composition at other positions of the C-domain, except for the AXA motif, may vary widely and has almost no effect on the interaction with SPase [66]. The amino acid residues located immediately after the cleavage site are referred to as the pro-region; its length can reach 30 aa. The first six residues of this region are the most important, being involved in the interaction with SPase [61,67]. The pro-region mainly consists of neutral and acidic residues. A substitution of negatively charged residues with basic ones deteriorates protein secretion and changes the cleavage site [53] (for SP sequences, see the Signal Peptide Website at http://www.signalpeptide.com (accessed on 25 September 2022), UniProt Database (accessed on 25 September 2022), and SPSED at http://www.spsed.com for their efficiency (accessed on 25 September 2022).

### 2.4. Using SPs Targeting to the ER and Its Derivatives to Raise the Accumulation Efficiency of Recombinant Proteins

The ER is the beginning of the secretory pathway for many proteins. Upon entering the ER, followed by a trip to the Golgi complex, proteins are subject to PTMs, folding, and quality control and can then be translocated to vacuoles or the apoplast with the help of COPII-coated vesicles [68]. Furthermore, plants have functionally specialized, ER-derived vesicles, including storage PBs [69,70]. Unlike COPII, ER-derived vesicles do not follow the standard route through the Golgi complex but are rather formed in the ER lumen, where the PBs either become independent storage entities or find themselves in storage vacuoles [71].

#### 2.4.1. Retention of Recombinant Proteins in the ER

Many therapeutic proteins, such as monoclonal antibodies or various enzymes, require correct glycosylation to preserve their pharmacological properties and optimal pharmacodynamics, as well as to guarantee the absence of adverse effects. Complex N-glycans, considerably differing between plants and mammals, are synthesized in the Golgi cis- and trans-cisternae. In plants, they contain xylose and fucose residues, which are absent in animal glycans. It is believed that these glycans can induce allergic reactions. However, the high-mannose N-glycans formed in the ER at initial glycosylation stages are identical between plants and humans [72]; thus, it is reasonable to prevent the protein translocation to Golgi cisternae by accumulating it in the ER instead. For this purpose, a short C-terminal sequence, (SE)KDEL/HDEL, acting as a signal for retrieving the synthesized protein to the ER, is attached to the protein in addition to the N-terminal SP [73,74]. Along with the prevention of undesirable glycosylation of the plant type, the retention of the protein in the ER enhances its accumulation because the ER contains only a small number of protease types [75]. Despite the concerns that high-mannose glycans have a negative effect on the pharmacological polypeptides of synthesized recombinant proteins, the method of protein retention in the ER is widely used because it allows for a rapid increase in the yield of a target protein [73]. The better accumulation and improved stability of many recombinant proteins in the cell have been shown for many species of transgenic plants as a result of targeting and retention in the ER [76,77,78,79,80,81]. Unfortunately, this translocation of recombinant proteins fails in certain tissues sometimes and has undesirable consequences. In particular, some recombinant antibodies targeted to the ER in *Arabidopsis* and tobacco seeds partially enter the Golgi complex, followed by the storage vacuoles and the apoplast, where they undergo glycosylation of an undesirable type or are degraded proteolytically [82,83,84,85].

#### 2.4.2. Targeting of Recombinant Proteins to PB-like Structures

Cells of the cereal seed endosperm possess two types of structures containing storage proteins, namely, ER-derived PBs (mainly composed of prolamins) and post-Golgi protein storage vacuoles (containing globulins and glutelins). Both organelles have a low amount of water and a low protease activity; accordingly, they are most suitable for the accumulation and long-term storage of recombinant proteins [73,86,87]. One of the PB types in the maize (*Zea mays*) endosperm is formed from proteins called zeins, which account for over half of the total endosperm protein [88]. The 27 kDa γ-zein is localized to the periphery of PBs formed in the ER, where it encompasses the aggregates of other zeins, stabilizes them, and enhances their sequential assembly into PBs [89,90]. Heterologous γ-zein expression in the leaf epidermis of *Arabidopsis* and tobacco leads to the formation of membrane-encompassed structures closely resembling PBs of the cereal endosperm, suggesting that one or several γ-zein structural motifs are responsible for the de novo formation of these bodies [91]. γ-Zein contains a proline-rich domain at its N terminus; the domain is represented by the PPPVHL sequence repeated eight times. An artificial peptide, Zera, is used to assemble the PBs of a target recombinant protein. This polypeptide comprises 112 aa and contains the signal γ-zein peptide (which targets it to the ER) and its proline-rich domain [92,93] (with six cysteine residues capable of forming disulfide bonds between chains, thereby ensuring the oligomerization of Zera-containing molecules and the formation of PBs) [94,95]. The sequence coding for the C-terminal end of the Zera peptide is directly fused to the gene of a target protein. The expression of this hybrid protein leads to PB formation not only in vegetative plant tissues but also in insect and mammalian cells [92,93]. The isolation and purification of the proteins fused to Zera facilitate the centrifugation and separation of the induced PBs in density gradients [95,96]. A number of therapeutic proteins, such as calcitonin, epidermal growth factor, and human growth hormone, have been successfully synthesized in plant cells via the fusion to Zera and PB induction [93]. A protein recombinant vaccine against human papillomavirus, with its Zera-fused E7 protein as the major component, has been successfully synthesized in *Nicotiana benthamiana* with a high yield and specific immunogenicity [97].

Along with Zera, two more peptides are successfully used for the translocation of target proteins to induced PBs; these peptides are elastin-like polypeptides (ELPs) and hydrophobins (HFBs). ELPs are synthetic peptides comprising pentapeptide VPGXG (X: any nonproline amino acid) repeated 5 to 160 times, first discovered in the elastin of mammals [98]. ELPs share structural characteristics with intrinsically disordered proteins and are capable of reversible transition from a soluble protein to insoluble aggregates and back depending on specific transition temperatures [99]. This property of ELPs can be exploited for the rapid purification of a protein fused to an ELP by an inverse transition cycling procedure [98]. The ELP of 30–40 VPGXG repeats considerably increases the accumulation of spider silk proteins, interleukins, and monoclonal antibodies such as PBs in the tobacco leaves and cells [100,101,102]. A GFP–ELP fusion considerably increases GFP (green fluorescent protein) accumulation in tobacco leaves (to 40% of the total soluble protein, TSP) because of the induction of the formation of GFP-containing PBs [103]. HFBs are a family of small secreted proteins synthesized by filamentous fungi—in particular, *Trichoderma reesei* [104]. HFBs are globular proteins with four disulfide bonds and a hydrophobic layer on the surface, which underlie their hydrophobic and surfactant characteristics. Proteins fused to HFBs acquire hydrophobic properties and can be utilized to purify the former via aqueous two-phase separation [105]. The transient expression of GFP fused to an HFB in *N. benthamiana* leaves gives rise to numerous PBs and increases GFP accumulation up to 51% of TSP. The fusion of other target proteins to HFBs also elevates their yield [106].

Of note, all three peptide agents induce PB assembly not only in plants but also in the cells of all three eukaryotic kingdoms, implying the high conservation of the mechanism underlying PB formation. The induced PBs coated with a membrane are protected from proteolytic degradation by cytosolic enzymes, while the cell itself is protected from undesirable effects of foreign proteins. In addition, PBs have a high density and are easily separable by centrifugation, which considerably facilitates the purification of target proteins. Thus, these properties are suggestive of the utility of induced PBs for the production of recombinant proteins.

#### 2.4.3. Targeting of Recombinant Proteins to Vacuoles

In seeds and other plant storage tissues, vacuoles are specialized for long-term protein storage; accordingly, the targeted translocation of recombinant proteins to vacuoles seems worthwhile [107]. On the contrary, the accumulation in vacuoles of leaves or undifferentiated cells in suspension culture is rather undesirable for many recombinant proteins because the conditions in these compartments cannot guarantee their stability. Nonetheless, there are some examples of a high accumulation of certain proteins (endolysin, avidin, and cellulolytic enzymes) in the central vacuoles of leaves [108]. The targeting of monoclonal antibodies to vacuoles has also been successfully implemented in *N. benthamiana* [109]. DLLVDTM, a vacuolar SP from chitinase A of tobacco, has been used to synthesize human β-glucocerebrosidase, a therapeutic for Gaucher’s disease. The glucocerebrosidase synthesized in carrot cell culture is the first pharmaceutical produced in a plant and approved for human use. To be pharmacologically active, this enzyme has to be glycosylated and contain exposed mannose residues. This particular type of glycosylation takes place in plant vacuoles and yields paucimannosidic-type N-glycan structures [110]. These specific features of the PTMs in vacuoles make them preferable for the translocation of some recombinant proteins. *Aspergillus niger* polygalacturonase, transiently expressed in *N. benthamiana* and targeted to vacuoles, accumulates there in a lower amount as compared with the enzyme targeted to the ER and apoplast but manifests a considerably higher activity, which is most likely explained by the specific PTM features in the vacuole [111]. One of the *Triticum aestivum* enzymes involved in fructan synthesis, 6-SFT, carries a 26-aa SP (at its N terminus) that targets it to vacuoles. GFP has been fused to this SP to obtain transgenic sugarcane plants with the vacuoles of stem parenchyma cells containing a large amount of GFP. A similar picture is observed for transient SP–GFP expression in celery and maize stigmata [112]. Consequently, the SPs targeting recombinant proteins to vacuoles can be effective at increasing the yield of a target product, especially if these proteins require specific PTMs implementable in vacuoles.

#### 2.4.4. Targeting of Recombinant Proteins to the Apoplast

Most of secreted proteins are transported via the route conserved among all eukaryotes, which starts from the translocation to the ER lumen through Golgi cisternae to be then delivered by vesicles to the cytoplasmic membrane and—with the help of exocytosis—to the apoplast [113,114]. To enter the ER, where a protein is folded and subjected to initial PTMs, the protein requires a specialized secretory SP at its N terminus [115,116]. Proteins are further post-translationally modified in the trans-cisternae, and some of them are translocated to vacuoles [117]. The apoplast targeting of recombinant proteins is widely used in the cultivation of transgenic plant cells because a protein of interest in this case is directly released into the culture medium to be subsequently extracted easily [118]. Most of secreted proteins carry a short (20–40 aa) peptide targeting them to the outside of the cell. The SPs of this type have no conserved regions, and individual secreted proteins may carry completely different SPs. On the other hand, the secretion mechanisms, which are highly conserved among eukaryotes, allow SPs of a heterologous origin—for example, animal SPs in a plant expression system—to be employed for targeting recombinant proteins to the apoplast. In particular, the authentic SP of human growth hormone ensures the correct processing and translocation of human growth hormone to the apoplast of *N. benthamiana* [16,119], as does the authentic SP of human acid α-glucosidase in *A. thaliana* cell culture [120]. Various SPs yield different secretion efficiency levels; hence, these peptides should be individually selected for each expression system. For example, the 33KDsp peptide has emerged as the most efficient in the rice suspension-cultured cells of three tested variants: it provides the targeting of up to 92% of a recombinant protein to the apoplast [121].

## 3. Protein Translocation to Endosymbiotic Organelles

Mitochondria and plastids are endosymbiotic organelles originating from ancestors of extant α-proteobacteria and cyanobacteria. The symbiosis commenced at different time points; the progenitors of mitochondria were the first to be taken up by a proto-eukaryotic cell, followed by plastids [122]. The transformation of endosymbionts into organelles has been accompanied by the massive translocation of their genetic material to the nucleus. Given that most of mitochondrial and plastid proteins are synthesized in the cytosol on nuclear transcripts [123,124,125], these proteins must be targeted from the cytosol to the corresponding organelles. The majority of proteins targeted to plastids and mitochondria carry at their N terminus an SP named the “transit peptide” for chloroplasts and “presequence” for mitochondrial targeting; it is recognized by import mechanisms of these organelles and then cleaved by specific signal peptidases. One of the most intriguing aspects is that the protein-targeting mechanisms of plastids and mitochondria are very similar [126]. Along with the specific sequences targeting proteins to either plastids or mitochondria, some SPs are recognizable by the import machineries of both organelles [127]. Recently, an ever-increasing volume of data revealed that proteins are targeted to mitochondria or to the ER not only by means of SPs but also at the level of the targeted transport of the mRNAs carrying zip codes for the interaction with RNA-binding proteins at their 3′ end [128,129]. As a result, the mRNAs and even mRNA complexes with ribosomes are localized near the mitochondrial outer membrane and even on its surface, thereby directly interacting with its import machinery. Note that protein targeting to mitochondria can be not only post-translational, as previously believed, but also cotranslational. The role of mRNA targeting in the case of plastids is rather vague; however, some indirect evidence suggests that it occurs there too [130,131]. This issue is beyond the scope of our review.

### 3.1. Major Mechanisms Underlying Protein Import into Mitochondria

Most proteins targeted from the cytosol to mitochondria carry an N-terminal SP referred to as presequence, which is subject to cleavage later. The main part of these precursor proteins is transferred to the translocon on the external side of the outer mitochondrial membrane (TOM complex, comprising several proteins), which is the main portal for the protein import into a mitochondrion [123,132]. Tom20 and Tom22 recognize the presequence of a transported precursor and directly bind to it, allowing for its passage through the pore formed by Tom40 to the intermembrane space of a mitochondrion. Small proteins, Tom5–Tom7, are involved in the functioning of the TOM complex [133]. Then, the presequence of the precursor protein binds to Tim50 of the TIM transmembrane complex (of the inner mitochondria membrane), which sorts the precursors and targets them to either the matrix or inner membrane or leaves them in the intermembrane space [132,134]. The matrix-targeted precursors pass through the pore formed by TIM23 coupled with an ATP-dependent translocon-associated motor. Upon entering the stroma, the presequences of precursors are cleaved by mitochondrial processing peptidase, and the protein is folded by the chaperones Hsp60 and Hsp10 [132].

Most of the mitochondrial proteins synthesized on eukaryotic 80S ribosomes are post-translationally transferred to these organelles [135,136]. However, the mechanisms of this protein traffic are studied insufficiently. It is known that molecular chaperons, such as Hsp70 and Hsp90, and their cochaperones play important roles in this process. Hsp70 binds to the short hydrophobic presequence of the protein, which explains its broad substrate specificity. Hsp70 maintains the protein in an unfolded state, which is necessary for its passing through the TOM complex to the intermembrane space [135,136]. Tom70 associated with Tom40 contains a TPR (tetratricopeptide repeat): the domain recruiting and binding molecular chaperones and preventing the aggregation of precursor proteins [137]. It is not clear how Hsp70 specifically mediates the targeting of mitochondrial proteins because the same chaperone targets proteins to the ER as well [138] (Figure 3A).

Hsp70 cochaperones and J proteins also take part in the transport of precursor proteins to mitochondria. The latter stimulate the Hsp70 ATPase activity, contain the domain-binding precursors, and are able to deliver them to Hsp70 [139,140]. Some precursors of hydrophobic proteins of the inner mitochondria membrane are first delivered to the outer ER surface but do not get integrated into the membrane. After that, one of the J proteins transfers these precursors from the ER surface to the outer mitochondrial membrane. This is the so-called ER–SURF pathway [141].

Some data suggest that the protein import into a mitochondrion follows the cotranslational pattern. First, translation-arrested ribosomes were observed by cryo-tomography on the mitochondrial surface in the region of TOM [142]. Second, many proteins of the inner mitochondrial membrane were shown to be synthesized on mitochondria-associated ribosomes [143]. Third, an mRNA coding for mitochondrial proteins was observed on the surface of mitochondria [144,145]. This mitochondrial localization of mRNA is associated with the presence of zip codes in its 3′ untranslated region and an interaction with the Puf3 mRNA-binding protein [146]. The transport of mRNAs of mitochondrial proteins, along with translating ribosomes to the surface of mitochondria, is not well understood. The cleaved presequence at the precursor’s N terminus is the first to leave the ribosome exit tunnel and to be able to interact with Tom20, thereby initiating the import of the nascent precursor [129,147]. The initiation of cotranslational import into a mitochondrion requires that the complex of a ribosome, mRNA, and nascent polypeptide chain (RNC) be in direct proximity to the TOM complex. Om14, an outer mitochondrial membrane protein, can serve as a receptor for RNC-associated complexes of this type [148,149]. The molecular mechanism behind the translocation of a nascent protein chain of the overall RNC complex from Om14 to the TOM complex has yet to be studied.

### 3.2. Main Mechanisms of Protein Import into Plastids

Most of the proteins transferred to plastids carry an SP (or transit peptide, determining their target) at their N terminus. The transit peptides targeting proteins to plastids are most different in their primary structure and length (30–150 aa) [150]. As a rule, transit peptides are cleaved from a proprotein in the plastid stroma by stromal processing peptidases (SPPs) [151]. Most proteins are transported to plastids in a post-translational manner and enter the TOC/TIC protein translocation system (Figure 3). The TOC translocon resides on the outer plastid membrane, while the TIC is responsible for the translocation across the inner membrane [152,153]. Numerous chaperone complexes of different families, including Hsp70 and Hsp90, interact with the proteins targeted to plastids and interfere with their folding, misfolding, and aggregation [154]. The transit peptides of plastid preproteins bind to the core of the TOC complex assembled from Toc34, Toc159, and Toc75 [155]. Toc34 and Toc159 are membrane-bound GTPases acting as primary import receptors of the outer membrane [156]. Their GTPase activity triggers the transit of a TOC-bound preprotein [157]. In this process, Toc75 is the major component forming the protein import channel of the outer membrane [158].

The proteins targeted to the plastid stroma have to pass through the inner membrane too. For this reason, the TOC complex is tightly associated with the second translocation system situated in the inner membrane, the TIC, forming a supercomplex for transferring proteins from the cytosol to the plastid stroma [159]. The joint work of the TOC and TIC complexes is coordinated by Tic236, which mediates physical contacts among the TOC, the TIC membrane channel, and components of the ATP-dependent molecular motor responsible for the preprotein translocation across two membranes of the plastid envelope [160]. Small Tic22 chaperons are involved in the transit of preproteins across the intermembrane space [161]. Tic20 and Tic110 form the main transmembrane channels across the inner membrane [162]. In the stroma, the imported protein interacts with the complex of stromal chaperons, comprising Hsp90C, cpHsp70, and ClpC/Hsp93, which provide the folding of the imported protein and its interaction with the stromal processing peptidase cleaving the transit sequence [151,163] (Figure 3B).

The main protein import routes to plastids are studied much worse as compared with mitochondrial ones; however, it is clear that the post-translational mechanism is not the only one. Proteins are also imported into plastids in a cotranslational manner via the targeting of plastid protein mRNAs in a complex with cytosol ribosomes onto the external surface of the plastid outer membrane. As shown by the in situ hybridization of *Chlamydomonas reinhardtii*, the mRNA coding for the LHCII chloroplast protein mainly accumulates in its basal region. Puromycin treatment interferes with this localization of the LHCII mRNA on the chloroplast membrane. It is noteworthy that puromycin causes a premature release of the nascent polypeptide chain from the ribosome; accordingly, the LHCII import into the chloroplast is explainable by the targeted transport of mRNA to the chloroplast membrane coupled with translation, that is, in a cotranslational manner [164]. Recent studies based on high-resolution electron tomography indicate that the cytosol ribosomes associated with the mRNAs coding for subunits of chloroplast proteins called LHCs (light-harvesting complexes) and RBCs (Rubisco small subunits) reside on the external surface of the basal part of the *C. reinhardtii* outer chloroplast membrane and are translationally active. Therefore, these proteins are imported into the chloroplast via TOC/TIC translocons in a cotranslational manner. Once cytosol ribosomes emerge on the chloroplast membrane, the membrane must have receptors for these ribosomes; the fact that their dissociation requires high ionic strength supports this theory [165].

Some proteins transferred to plastids require glycosylation for their functionality. At their N terminus, they carry a peptide targeting them to the ER. They are then imported in a cotranslational manner to go to the Golgi complex, where they are glycosylated before reaching the target plastids. The mechanism underlying the import from the Golgi complex is still obscure [166,167].

### 3.3. Structure of SPs Targeting Proteins to Endosymbiotic Organelles

The analysis of the structure of many transit peptides in chloroplast proteins and presequences of mitochondrial proteins suggests that their signal sequences are rather similar in their amino acid composition despite tremendous differences in primary structures [168]. Neither the former nor latter contain any conserved consensus sequences. The differences between various transit peptides and presequences are the same as those between these groups of SPs. The analysis of the primary structure of transit peptides and presequences indicates that they contain numerous dispersed short motifs responsible for different stages of import into the respective organelles (cytosol navigation, interaction with TOC/TOM and TIC/TIM complexes, and interaction with chaperones) [169,170] (Figure 4A).

Two functional domains are distinguishable in the transit peptides and presequences, namely, an N-terminal specificity domain (NSD) and a C-terminal translocation domain (CTD). Although the transit peptides and presequences are very similar, they have a difference in their structure: the N-terminal region of transit peptides is mostly hydrophobic and forms random coils, in contrast to presequences, where it is less hydrophobic, contains multiple arginine residues (MAR) and the moderately hydrophobic sequence motif (MHSM), and forms an amphiphilic α-helix [169,171]. The charge and amino acid composition of the N-terminal domain are determinants of the target (chloroplast, mitochondrion, or both organelles) of a signal sequence [127,171,172]. The removal of the MAR from a presequence switches the targeting from mitochondria to chloroplasts [169]; the addition of the MAR to the NSD of transit peptides blocks the targeting to chloroplasts, and the protein remains in the cytosol; an extra copy of the MHSM retargets the protein to mitochondria. Nevertheless, the addition of the MHSM alone fails to retarget the transit peptide to chloroplasts [169,173]. Thus, the presence or absence of MAR in SPs is the key determinant of the targeting to mitochondria or chloroplasts. The C-terminal domain (CTD) of transit peptides and presequences is the signal for the transit through TOC/TIC and TOM/TIM import systems [19,169,174] (Figure 4B). The CTDs of both systems of transmembrane translocation are almost interchangeable, i.e., the mitochondrial system recognizes a chloroplast CTD signal and vice versa [169]. See the UniProt Database (https://www.uniprot.org, accessed on 25 September 2022) for the sequences of chloroplast transit peptides and presequences of mitochondria.

Consequently, the transit peptides and presequences targeting proteins to plastids and mitochondria, respectively, are rather alike and share similar structures but lack any consensus sequences. A multitude of SPs are capable of concomitant targeting to both organelles, i.e., dual targeting. These signal sequences guide the proteins functioning in both compartments [169,175,176]. Accordingly, the question arises of whether and to what degree the presequences are conserved among eukaryotes and whether plant presequences differ from animal and fungal ones, which do not need any segregation between mitochondrial and plastid targeting. Amazingly, the presequences of animal and fungal mitochondrial proteins are capable of mitochondrial targeting in plant cells, whereas plant presequences can target to animal mitochondria. The replacement of the MAR with alanine residues in a nonplant presequence retargets the protein to plant cell chloroplasts. Moreover, MAR and MHSM insertion into an NSD of a plant transit peptide guarantees the protein’s targeting to mitochondria in animal cells [177]. Most likely, the need for transit signals appeared at the early stages of organellogenesis, when the nucleus started capturing the genetic material of proto-organelles. It is unlikely that this signal mechanism arose de novo; rather, one of the pre-existing cell systems formed its basis. According to one of the hypotheses, the NSD of SPs of plastids and mitochondria has originated from the bacterial twin-arginine translocation (TAT) SP, which can substitute for the presequence in both plant and animal cells [177]. Another hypothesis postulates that the SPs for protein targeting to endosymbiotic organelles have originated from the antimicrobial peptides of the host cell, thus destabilizing the cell membrane of prokaryotes, and have been imported by them for detoxification by proteolysis in the cytoplasm [178,179].

### 3.4. Targeting of Recombinant Proteins to Endosymbiotic Organelles

Plastids possess their own genome and are the most attractive for the biosynthesis of recombinant proteins because transplastomic plants have a number of advantages over nuclear transformants. However, the creation of homoplastomic plants or cell cultures entails certain difficulties, making the transformation of the nuclear genome a much easier task [180,181]. On the other hand, chloroplasts perhaps have the lowest level of proteolytic activity among all cell compartments [182] and are readily separable from the remaining cell components; therefore, they are perfectly suited for the storage of produced recombinant proteins.

For the targeting of recombinant proteins to the chloroplast, they are fused to a suitable N-terminal peptide—most often, to the transit peptide of the Rubisco small subunit (RbcS). This approach frequently exerts a considerable positive effect on the protein accumulation and is thus widely used, especially in the design of plant-made vaccines for humans and animals [183]. Over the last 30 years, more than 10 recombinant proteins synthesized from a nuclear transcript have been targeted to chloroplasts. The transit peptides of, e.g., RbcS, chlorophyll a/b-binding protein, and granule-bound starch synthase have been utilized for the targeting [184]. Such proteins as phosphoenolpyruvate synthetase, β-glucuronidase, and xylanase have been synthesized and accumulated in different plant species, such as petunia, potato, tobacco, and rice [185,186,187,188]. The level of protein accumulation has varied considerably. In particular, the resulting xylanase constitutes almost 5% of TSP, in contrast to phosphoenolpyruvate synthetase, which constitutes only 0.1% of TSP [185,188]. The amount of GFP synthesized in rice cells reaches 10% of TSP [189]. Of interest is that the RbcS transit protein has been used for the chloroplast targeting of all three proteins [184]. The Cry1Ac endotoxin of *Bacillus thuringiensis* and the Cel5A endoglucanase have been produced in rice and tobacco cells in an amount of 2% and 5% of TSP, respectively [190,191]. In *N. benthamiana*, the transient expression of fused p17/p24 HIV-1 (human immunodeficiency virus type 1) proteins carrying the RbcS transit peptide (ensuring chloroplast targeting) raises the expression eightfold (to ~4 mg/kg) as compared with the cytosol and ER localizations (~0.5 mg/kg) [192]. The yield of a human papillomavirus type 16 (HPV-16) capsid protein, L1, targeted to chloroplasts, reaches 11% of TSP in the case of nuclear expression and 17% in the case of transient expression [193]. As shown later, the L1 protein targeted to chloroplasts forms virus-like particles, enabling the production of a commercial virus-like-particle-based vaccine [194]. The HPV-16 oncoprotein, E7, when fused to an anti-lipopolysaccharide factor fragment (LALF32–51) transiently expressed in *N. benthamiana* leaves and targeted to chloroplasts, shows a 27-fold increase in its yield as compared with the cytosol localization [195].

The mitochondrial targeting of recombinant proteins is not as widely used for chloroplast targeting, except for the targeting of proteins, peptides, and other therapeutics and nanoparticles to mitochondria in the case of mitochondrial diseases, cancer, and many energy generation problems and other metabolic disorders [196,197]. This is a large research field beyond the scope of our review. Nonetheless, the mitochondrial targeting of recombinant proteins in plants makes sense for modulating certain functions of mitochondria, such as energy generation or responses to biotic and abiotic stressors. A low-oxygen mitochondrial environment is suitable for metabolic engineering based on oxygen-sensitive enzymes [198,199]. The mitochondrion is an isolated compartment surrounded by a double membrane and containing only a small number of proteases [182]. Baysal et al. [200] have studied the efficiency of six presequence peptides in the targeting to rice mitochondria. ATPA and COX4 (*Saccharomyces cerevisiae*), SU9 (*Neurospora crassa*), pFA (*A. thaliana*), and OsSCSb (*Oryza sativa*) successfully targeted the eGFP protein to mitochondria, whereas MTS2 (*Nicotiana plumbaginifolia*) almost completely failed despite its plant origin [200].

Nitrogenase cofactor maturase Nif comprises four polypeptide chains and serves as a cofactor of all nitrogenases (key enzymes in the bacterial nitrogen fixation pathway) of diazotrophic bacteria and archaea. Its components have been expressed in *N. benthamiana* leaves and successfully targeted to mitochondria with the help of COX4 (NifB) and SU9 (NifU, NifS, and FdxN) presequences. The attainment of high levels of a soluble and functional Nif cofactor in plant mitochondria is of paramount importance for the subsequent construction of a nitrogen fixation pathway in plants [201].

## 4. Prediction of Signal Sequences That Determine Protein Localization in the Cell

Specific intracellular sorting signals considerably differ in their sequence, structure, and length between individual proteins targeted to the same compartment and between proteins localized to different organelles. The N-terminal targeting peptides guiding proteins to the secretory route (SPs), mitochondria (presequences), chloroplasts (transit peptides), and specific compartments in mitochondria and chloroplasts are the most widespread. Because these signals target the protein transport in the cell, it is important that a researcher can precisely detect them in sequences of protein-coding genes. Accordingly, a large number of computer programs utilizing various machine learning algorithms have been designed over the last quarter century, including Grammatical Restrained Hidden Conditional Random Fields, support vector machines, N-to-1 extreme learning machines, Markov chains, profile-hidden Markov models, and neural networks [202,203,204,205,206,207]. See the review by Nielsen et al. [208] for a brief history of the creation and application of these methods for the prediction of the sorting signals in proteins. The very first software predicting signal sequences in proteins was SignalP 1.0, designed in 1996 and based on machine learning principles; its version 6.0 appeared in 2021 (https://services.healthtech.dtu.dk/service.php?SignalP-6.0 (accessed on 25 September 2022)) [209]. Unfortunately, SignalP only predicts SPs and is unable to predict presequences and transit peptides for mitochondrial and chloroplast targeting. Among the best-known software packages for the prediction of signal sequences is TargetP 2.0, based on neural networks. TargetP utilizes feed-forward networks and position weight matrices to analyze an amino acid sequence and predict SPs of the secretory route, mitochondrial presequences, chloroplast transit peptides, and thylakoid luminal transit peptides as well as the positions of their cleavage sites. TargetP predicts SPs with a probability of 97% and transit peptides and presequences with a probability of 90% [210]. TargetP 2.0 is available at https://services.healthtech.dtu.dk/service.php?TargetP-2.0 (accessed on 25 September 2022). The TargetP predictions of SPs match, with 90% agreement, the peptides annotated in the UniProt database, whereas its predictions of presequences and transit peptides yield only 80% agreement. Notably, TargetP 2.0 predicts twice as many mitochondrial proteins in plant proteomes as compared with animal proteomes. For grape and rice chloroplasts, this program outputs these numbers at 1125 and 2049, respectively [210]. Another useful software application for the prediction of the signal sequences of all targeting types in nine subcellular locations, DeepLoc 2.0, is based on convolutional neural networks (and is available at https://services.healthtech.dtu.dk/service.php?DeepLoc-2.0, accessed on 25 September 2022) [211]. The review by Jiang et al. [207] gives an impressive list of computer programs predicting SPs and shows the algorithms used, the types of signal sequences, and the server addresses.

## 5. Conclusions

The main problem with plant systems for the expression of recombinant proteins is a low yield of a target protein. This problem can be addressed via various approaches, including the translocation of a target protein to the cell compartments featuring a low protease activity. In addition, many therapeutic proteins require correct mammalian-type glycosylation for their functionality. All these problems can be solved by fusing target protein genes to signal sequences that determine the transport of the synthesized or nascent protein to certain cell compartments with either a low protease activity or appropriate glycosylation machinery. In their Golgi apparatus, plant cells form complex N-glycans with terminal fucose and xylose residues, absent in mammalian glycans. However, the initial glycosylation stages taking place in the ER and associated with the biosynthesis of polymannose glycans are similar between plants and animals. That is why it is worthwhile to target the recombinant therapeutic proteins requiring glycosylation to the ER with the help of an N-terminal SP and keep them there by means of a C-terminal ER retrieval peptide, thereby preventing their translocation to the Golgi complex. Furthermore, the protease localization in the ER is much better compared with the cytosol, thus also enhancing the accumulation of a target protein. If a recombinant protein requires glycosylation of the paucimannosidic type, it is reasonable to target it using an SP to vacuoles, where plants implement this type of complex glycosylation. In some cases, it is beneficial to target a recombinant protein to the apoplast (where the protease activity is weak) through the secretory route. Moreover, in the case of cultured cells, it is the culture medium that allows the target protein to be extracted rather easily.

In terms of the preservation and accumulation of recombinant proteins, the most attractive approach is to construct artificial PBs that are actually membrane-enveloped microvacuoles with a high content of the target protein. It is rather easy to generate such PBs by fusing the target gene to the sequences coding for either the N-terminal domain of γ-zein or elastin-like/hydrophobin peptides. Additionally, these PBs have a high density and are readily extractable by density gradient centrifugation. In the same way, the accumulation of target proteins in plastids, mainly chloroplasts, is the most promising because these organelles possess the lowest protease activity of all cell compartments and are separated from the cytosol by a double membrane. Plastids have their own genome and are transformable themselves; these properties also offer a number of advantages over nuclear transformation. Nonetheless, several difficulties arise there that are linked with the construction of homoplastidic and homoplastomic plant cells. Consequently, it is much easier to produce nuclear transformants and equip the protein of interest with a transit peptide targeting it to chloroplasts. Thus, the yield of the recombinant protein can be strongly increased via the purposeful use of translocation signal sequences. However, it should be made clear that different transport peptides have dissimilar efficiency levels; therefore, each recombinant protein requires an individual approach.

## Figures and Tables

**Figure 1 plants-11-02561-f001:**
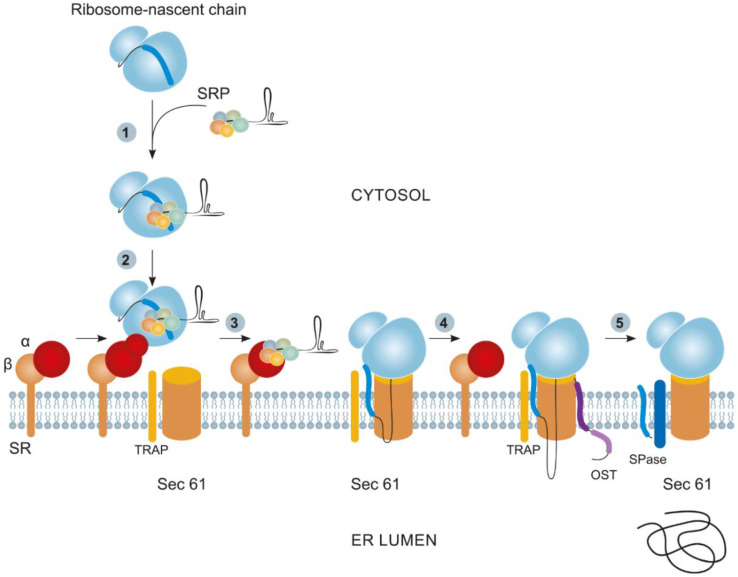
The cotranslational protein targeting pathway: (1) The SRP binds a signal sequence (dark blue) as it emerges from a ribosome to form the RNC–SRP complex; (2) the RNC–SRP complex docks with the ER membrane by binding to a cognate SRP receptor (SR, consisting of α and β subunits); (3) subsequently, the RNC is transferred from SR–SRP to translocon Sec61, resulting in the intercalation of the signal sequence into the translocon pore and its opening; (4) the nascent chain is translocated through the pore, and the SRP disengages from the SR; (5) concomitantly with translocation, the signal peptidase (SPase) and oligosaccharyltransferase (OST) enzyme complexes are recruited to the translocon to cleave the SP and add N-linked glycans to the nascent chain, respectively. The termination of the protein synthesis releases the nascent chain from the ribosome; the translocation is completed; and the protein folds in the ER lumen (adapted from [38] with permission).

**Figure 2 plants-11-02561-f002:**
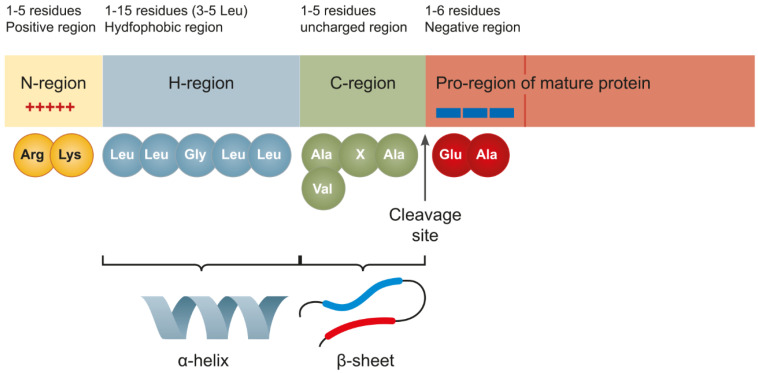
General structure of an SP. It is composed of three main parts: (1) an N-terminal region (a positively charged domain); (2) H-region (the hydrophobic core forming an α-helix); and (3) a C-terminal region (a cleavage site forming a β-sheet). The initial part of the protein, important to protein secretion, is referred to as the pro-region. Cleavage occurs within the AXA or VXA motif (adapted from [19], with permission).

**Figure 3 plants-11-02561-f003:**
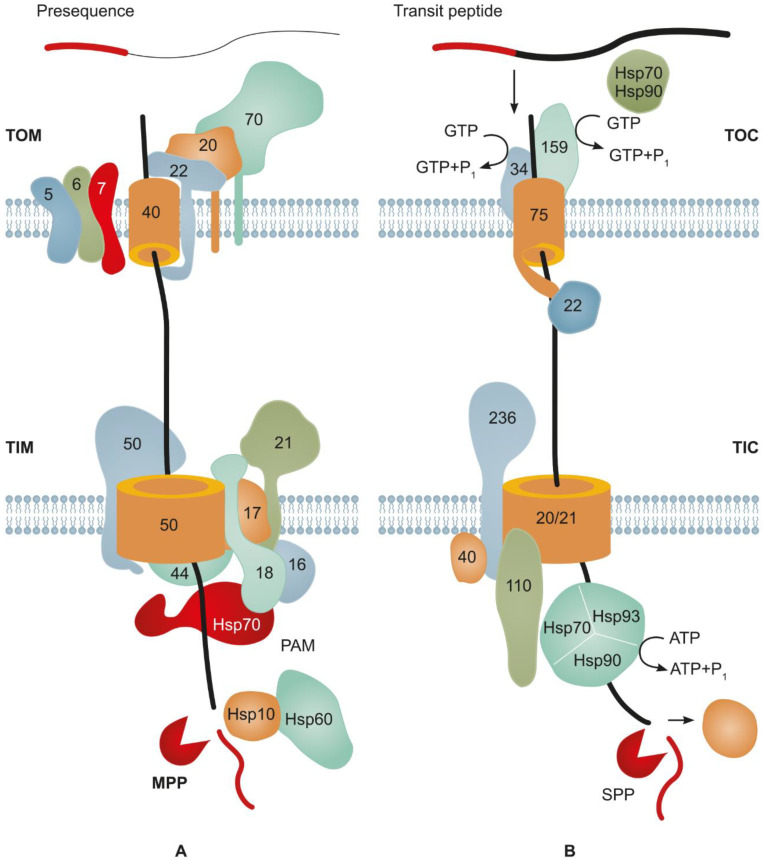
Mitochondrial and chloroplast protein import machineries. (**A**) Mitochondrial transmembrane protein transport. TOM, translocon of the outer mitochondrial membrane; TIM, translocase of the inner mitochondrial membrane; PAM, presequence translocon-associated motor; and MPP, mitochondrial processing peptidase. Approximate molecular weights (kDa) are indicated on individual TOM, TIM, and PAM components. See the main text for mechanistic details (adapted from [135], with permission). (**B**) Core components of TOC–TIC import machinery of chloroplasts. A newly synthesized preprotein is targeted to the TOC complex at the outer membrane by the binding of its transit peptide to receptors Toc34 and Toc159 with GTPase activities. The targeting is aided by the cytosolic chaperone complexes of Hsp70 and Hsp90. The GTPase activity of the receptors is required for the transport of the preprotein through the TOC and TIC transmembrane channels formed by TOC75 and TIC20/21, respectively. The TOC–TIC supercomplex assembled through the binding of Tic236 to Toc75 and of Tic110 to the TIC complex facilitates the direct transport of the preprotein from the cytosol to the chloroplast stroma. Tic110 and Tic40 (40), together with the chloroplast chaperones Hsp70, Hsp93, and Hsp90, form the ATP-dependent protein motor performing import into the chloroplast stroma. After the import, the transit peptide is cleaved by stromal processing peptidase (SPP) (adapted from [159], with permission).

**Figure 4 plants-11-02561-f004:**
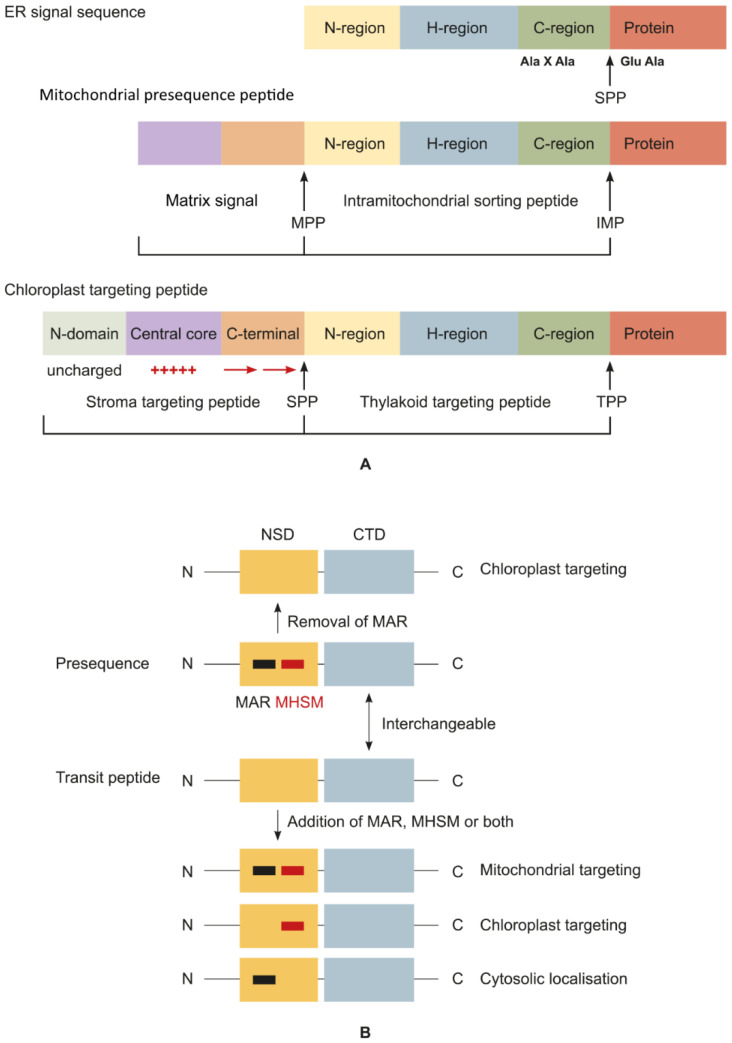
Specific features of the structure of mitochondrial and chloroplast SPs. (**A**) Eukaryotic signal sequences depending on the target organelle: the ER SP has a common tripartite structure; the mitochondrial-targeting peptide is composed of a matrix signal and intramitochondrial sorting signal; and the chloroplast transit peptide comprises the stroma-targeting peptide and thylakoid-targeting peptide. The intramitochondrial sorting signal and thylakoid-sorting signal share a common tripartite structure. Upward arrows denote the cleavage site, and the horizontal arrow denotes a β-sheet (adapted from [19], with permission). (**B**) The relation between the chloroplast transit peptide and mitochondrial presequence. The N-terminal specificity domain (NSD) of mitochondrial presequences contains multiple arginine residues (MAR) and a moderately hydrophobic sequence motif (MHSM). The removal of the MAR is sufficient to switch the targeting specificity from mitochondria to chloroplasts. Conversely, the incorporation of both the MAR and MHSM into the NSD of chloroplast transit peptides changes the targeting specificity from chloroplasts to mitochondria. The insertion of the MAR or MHSM alone into a transit peptide results in cytosolic localization or chloroplast targeting, respectively. C-terminal translocation domains (CTDs) are interchangeable between these targeting signals (adapted from [170], with permission).

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
