# Peer review of "Increasing the Efficiency of the Accumulation of Recombinant Proteins in Plant Cells: The Role of Transport Signal Peptides"

_plants, 2022, doi:10.3390/plants11192561_

Round 1
Reviewer 1 Report
The manuscript “Increasing the Efficiency of Accumulation of Recombinant Proteins in Plant Cells: The Role of Transport Signal Peptides” is based on an extensive literature review citing more than 200 studies conducted within this research field. This review highlights the role of transport signal peptides in plant science research, and gives insights the knowledge, applications, advantages and limitations of using such peptides to increase the accumulation of recombinant proteins in plant systems. The manuscript is well structured and illustrated with informative figures. I believe this manuscript is a timely review on this important technology and will therefore be of interest for a wide audience.
However, specific comments are listed below:
Major comments:
1. In the “Abstract” section, the authors use the term «translocation» (see - The problem of an increase the yield of recombinant proteins is resolvable using different approaches, including the translocation of a target protein to the cell compartments with a low protease activity), it should be noted that, as a rule, this term is used to refer to a type of chromosomal mutation in which a portion of a chromosome is transferred to a non-homologous chromosome. A different term should be chosen.
2. “Introduction” section. When the authors point out that “In the cytosol, proteins can undergo undesirable modifications altering their folding [13]; however, all improperly folded proteins are degraded according to ubiquitin proteasome proteolytic pathway [14]” Readers might take it this way: if degradation occurs only through misfolding, then it is not transport that is important, but ensuring the correct folding of the recombinant protein. An explanation is required of the role of recombinant protein transport in preventing its proteolytic degradation.
3. “Introduction” section. The authors indicate “These peptide signal sequences can be successfully used to target the recombinant proteins synthesized in plant cells to the compartments with a low protease activity”, and the question arises - is it known which compartments have low proteolytic activity or is this a further direction of research? Possibly a modern review - Philippe V Jutras, Isobel Dodds and Renier AL van der Hoorn. Proteases of Nicotiana benthamiana: an emerging battle for molecular farming. Current Opinion in Biotechnology 2020, 61:60–65. https://doi.org/10.1016/j.copbio.2019.10.006 - might be helpful to clarify this kind of issue.
4. “Intracellular Transport of Proteins to the ER and Its Derivatives” section. When the authors describe two types of protein transport (“Commonly, two types of protein transport to all eukaryotic cell compartments are distinguished, namely, cotranslational and posttranslational”), readers may find it interesting whether it is possible to predict which way the protein transport will go - co-translational or post-translational? It would be helpful to mention this in a clearer way.
5. Section - Using Signal Peptides Targeting to the ER and Its Derivatives to Increase the Accumulation Efficiency of Recombinant Proteins - it would be appropriate to give examples of signal peptides for protein transport in the ER, as is done for other plant cell compartments.
6. In my opinion, a table that presents examples of signal transport peptides that ensure the transport of recombinant proteins into different compartments of plant cells would be extremely useful for summarizing the material and for readers.
Minor comments:
1. It is more professional to use the term “N-terminus” instead of “N-end”, as well as the term “C-terminus” instead of “C-end”.

Author Response
Dear Reviewer,
Thank you very much for your critical reviewing of our work. We have studied all comments carefully and have made corrections which we hope meet with their approval. I will try to respond to your comments and suggestions.
Major comments:
- In the “Abstract” section, the authors use the term «translocation» (see - The problem of an increase the yield of recombinant proteins is resolvable using different approaches, including the translocation of a target protein to the cell compartments with a low protease activity), it should be noted that, as a rule, this term is used to refer to a type of chromosomal mutation in which a portion of a chromosome is transferred to a non-homologous chromosome. A different term should be chosen.
Fixed. Translocation changed to transport.
- “Introduction” section. When the authors point out that “In the cytosol, proteins can undergo undesirable modifications altering their folding [13]; however, all improperly folded proteins are degraded according to ubiquitin proteasome proteolytic pathway [14]” Readers might take it this way: if degradation occurs only through misfolding, then it is not transport that is important, but ensuring the correct folding of the recombinant protein. An explanation is required of the role of recombinant protein transport in preventing its proteolytic degradation.
Fixed. Cytosol has a high protease activity, moreover, in the cytosol, proteins can undergo undesirable modifications altering their folding [13]. However, all improperly folded proteins are degraded according to ubiquitin proteasome proteolytic pathway [14], which will lead to an even greater drop in the yield of recombinant protein.
- “Introduction” section. The authors indicate “These peptide signal sequences can be successfully used to target the recombinant proteins synthesized in plant cells to the compartments with a low protease activity”, and the question arises - is it known which compartments have low proteolytic activity or is this a further direction of research? Possibly a modern review - Philippe V Jutras, Isobel Dodds and Renier AL van der Hoorn. Proteases of Nicotiana benthamiana: an emerging battle for molecular farming. Current Opinion in Biotechnology 2020, 61:60–65. https://doi.org/10.1016/j.copbio.2019.10.006 - might be helpful to clarify this kind of issue.
Unfortunately, this article does not consider the number of proteases in different cell compartments, but only the number of different types of proteases in the total proteome. And it is very difficult to compare protease activity in different cell compartments, since different compartments contain predominantly different types of proteases, and different proteins have different sensitivity to different proteases.
Cell organelles contain different types and amounts of proteases, the lowest protease activity is in apoplast and plastids [17,18], and various recombinant proteins are to different degrees prone to degradation by individual proteases. Correspondingly, the selection of a cell compartment for accumulation of a particular recombinant protein is not universal.
- “Intracellular Transport of Proteins to the ER and Its Derivatives” section. When the authors describe two types of protein transport (“Commonly, two types of protein transport to all eukaryotic cell compartments are distinguished, namely, cotranslational and posttranslational”), readers may find it interesting whether it is possible to predict which way the protein transport will go - co-translational or post-translational? It would be helpful to mention this in a clearer way.
Fixed.
Commonly, two types of protein transport to all eukaryotic cell compartments are distinguished, namely, cotranslational and posttranslational. In the former, protein is synthesized directly on the membrane of the ER or an organelle and cotranslationally transferred through the membrane and in the latter, a protein completely synthesized in the cytosol free ribosomes is delivered one way or the other to its target residence. It is not entirely clear what determines the choice of one or another way.
- Section - Using Signal Peptides Targeting to the ER and Its Derivatives to Increase the Accumulation Efficiency of Recombinant Proteins - it would be appropriate to give examples of signal peptides for protein transport in the ER, as is done for other plant cell compartments.
Signal peptides are very diverse and do not have conserved consensus sequences, only a single general structure plan, which is described in the review. With regard to specific signal sequences, we believe that there is no need to give them in the review, since we are referring the reader to extensive databases.
- In my opinion, a table that presents examples of signal transport peptides that ensure the transport of recombinant proteins into different compartments of plant cells would be extremely useful for summarizing the material and for readers.
We considered the possibility of creating such a table, but came to the conclusion that it would be more correct to abandon it. Since Signal peptides are very diverse and do not have conserved consensus sequences, only a single general structure plan, there is no point in listing several randomly selected peptide sequences.
Minor comments:
- It is more professional to use the term “N-terminus” instead of “N-end”, as well as the term “C-terminus” instead of “C-end”.
Fixed.
Best regards,
Sergey Rozov

Reviewer 2 Report
The revision presented by Sergey M. Rozov and Elena V. Deineko focuses on the accumulation of recombinant proteins in plant cells, by targeting them to specific organelles. The manuscript depicts very well the data available in the literature and compartmentalizes the information into several subchapters making the text flow more easily. Nevertheless, the entire manuscript needs extensive English revision, especially regarding sentence construction. Other minor questions that in my opinion should be addressed are listed below:
· The authors mention the importance of post-translational modifications and their role in the correct function of the protein. Regarding glycosylation, one of the most relevant modifications for mammalian proteins, the authors state, correctly, that glycosylation acquired in the ER of plant cells is similar to the one of mammalian proteins. However, the authors fail to explain how the glycosylation trimming in the plant Golgi is substantially different from the mammalian counterpart. This issue is mentioned briefly in the conclusion of the manuscript, but I believe that it should be addressed and explained when the authors talk about glycosylation. Moreover, The authors mention that there is a glycosylation step occurring in the vacuole, which, to my knowledge, is not correct.
· The manuscript is presented with three different figures. In my opinion, apart from the second one, the figures do not add novel information to the manuscript as refer to specific events mentioned in the review but do not cover all the processes/pathways described. If the authors wish to include a figure, which I fully support, I believe that a summary of all the processes or pathways described in the manuscript would be more informative.
· All the manuscript need a revision regarding the formatting of the text and figures and, in the reference section, all items should be formatted equally (for example, for some references the DOI is included and for others, it is not).
Based on my analysis of the manuscript I recommend it to be revised based on the points I described before and after a careful revision of the English, which, in my opinion, would greatly improve the quality of the work.
Author Response
Dear Reviewer,
Thank you very much for your critical reviewing of our work. We have studied all comments carefully and have made corrections which we hope meet with their approval. I will try to respond to your comments and suggestions.
The revision presented by Sergey M. Rozov and Elena V. Deineko focuses on the accumulation of recombinant proteins in plant cells, by targeting them to specific organelles. The manuscript depicts very well the data available in the literature and compartmentalizes the information into several subchapters making the text flow more easily. Nevertheless, the entire manuscript needs extensive English revision, especially regarding sentence construction.
The English language was corrected and certified by native speakers from shevchuk-editing.com.
- The authors mention the importance of post-translational modifications and their role in the correct function of the protein. Regarding glycosylation, one of the most relevant modifications for mammalian proteins, the authors state, correctly, that glycosylation acquired in the ER of plant cells is similar to the one of mammalian proteins. However, the authors fail to explain how the glycosylation trimming in the plant Golgi is substantially different from the mammalian counterpart. This issue is mentioned briefly in the conclusion of the manuscript, but I believe that it should be addressed and explained when the authors talk about glycosylation.
The difference between mammalian and plant Golgi glycosylation is described in the text chapter 2.4.1. :
. Complex N-glycans, considerably differing between plants and mammals, are synthesized in the Golgi cis- and trans-cisternae. In plants, they contain xylose and fucose residues, which are absent in animal glycans. It is believed that these glycans can induce allergic reactions.
Moreover, The authors mention that there is a glycosylation step occurring in the vacuole, which, to my knowledge, is not correct.
The main part of vacuolar glycoproteins lacks terminal La epitops and terminal GlcNAc residues, forming oligomannose glycans (Gomord, V., Fitchette, A.C., Menu-Bouaouiche, L., Saint-Jore-Dupas, C., Plasson, C., Michaud, D., Faye, L. (2010) Plant-specific glycosylation patterns in the context of therapeutic protein production, Plant Biotechnol J., 8, 564–587.) and paucimannose glycans (Shaaltiel, Y.; Bartfeld, D.; Hashmueli, S.; Baum, G.; Brill-Almon, E.; Galili, G.; Dym, O.; Boldin-Adamsky, S.A.; Silman, I.; Sussman, J.L.; et al. Production of glucocerebrosidase with terminal mannose glycans for enzyme replacement therapy of Gaucher’s disease using a plant cell system. Plant Biotechnol. J. 2007, 5, 579–590.)
- The manuscript is presented with three different figures. In my opinion, apart from the second one, the figures do not add novel information to the manuscript as refer to specific events mentioned in the review but do not cover all the processes/pathways described. If the authors wish to include a figure, which I fully support, I believe that a summary of all the processes or pathways described in the manuscript would be more informative.
The main goal of this review was to consider the possibility of increasing the yield of recombinant proteins by transporting them to various cell compartments using transport peptides. Therefore, a detailed consideration of all possible ways of transporting proteins through membranes was not included in our tasks. We have only focused on the main pathways by which recombinant proteins can be transported to different compartments of the cell. Consideration of all possible ways is beyond the scope of this review. Therefore, in the figures, we have shown only the main transport machines that can be used for the transport of recombinant proteins of interest, and the main structural features of the transport peptides. Without these figures, it would be difficult for the reader to understand the text of the review.
- All the manuscript need a revision regarding the formatting of the text and figures and, in the reference section, all items should be formatted equally (for example, for some references the DOI is included and for others, it is not).
Based on my analysis of the manuscript I recommend it to be revised based on the points I described before and after a careful revision of the English, which, in my opinion, would greatly improve the quality of the work.
English language of the review was revised with the help of the native speakers from shevchuck-editing.com, and all references are formatted equally – now all references contain DOI addresses.
Thank you one more time for your critical reviewing of our work,
Best regards,
Serge Rozov

Round 2
Reviewer 2 Report
The revised version of the manuscript “Increasing the Efficiency of Accumulation of Recombinant Proteins in Plant Cells: The Role of Transport Signal Peptides” presented by Sergey M. Rozov and Elena V. Deineko represents a great improvement from the previously submitted version. The authors did a great effort in improving the English, which is now quite good, and addressed all the comments and concerns raised, by altering the manuscript or providing a sound explanation.
Based on my new analysis of the manuscript I recommend it be published in its present form.